# The Role of Mitophagy in Viral Infection

**DOI:** 10.3390/cells11040711

**Published:** 2022-02-17

**Authors:** Yuwan Li, Keke Wu, Sen Zeng, Linke Zou, Xiaowen Li, Chen Xu, Bingke Li, Xiaodi Liu, Zhaoyao Li, Wenhui Zhu, Shuangqi Fan, Jinding Chen

**Affiliations:** 1College of Veterinary Medicine, South China Agricultural University, No. 483 Wushan Road, Tianhe District, Guangzhou 510642, China; waner20191028012@stu.scau.edu.cn (Y.L.); 13660662837@163.com (K.W.); 13425951792@163.com (S.Z.); zlk13592585967@163.com (L.Z.); 18306616234@163.com (X.L.); 97xc@stu.scau.edu.cn (C.X.); 18233523828@163.com (B.L.); lxd18839462378@163.com (X.L.); lzhaoyao123@163.com (Z.L.); 13970064482@163.com (W.Z.); 2Guangdong Laboratory for Lingnan Modern Agriculture, Guangzhou 510642, China; 3Key Laboratory of Zoonosis Prevention and Control of Guangdong Province, Guangzhou 510642, China

**Keywords:** mitophagy, viral infection, mechanism, mitophagy receptors

## Abstract

Mitophagy, which is able to selectively clear excess or damaged mitochondria, plays a vital role in the quality control of mitochondria and the maintenance of normal mitochondrial functions in eukaryotic cells. Mitophagy is involved in many physiological and pathological processes, including apoptosis, innate immunity, inflammation, cell differentiation, signal transduction, and metabolism. Viral infections cause physical dysfunction and thus pose a significant threat to public health. An accumulating body of evidence reveals that some viruses hijack mitophagy to enable immune escape and self-replication. In this review, we systematically summarize the pathway of mitophagy initiation and discuss the functions and mechanisms of mitophagy in infection with classical swine fever virus and other specific viruses, with the aim of providing a theoretical basis for the prevention and control of related diseases.

## 1. Introduction

### 1.1. Mitochondria and Mitophagy

Mitochondria are organelles that produce energy in cells and are the primary site of aerobic respiration, which is why they are called “powerhouses”. Mitochondria participate in various crucial cellular processes, such as ATP production, apoptosis, calcium homeostasis, cellular proliferation, and the synthesis of amino acids, nucleotides, and lipids. Maintenance of the standard shape and function of mitochondria is an essential process for ensuring that cells perform various physiological activities. Upon exposure to stress, such as oxidative stress, mitochondria are prone to damage and dysfunction. The process through which excess or damaged mitochondria are selectively eliminated is called mitophagy. Mitophagy is categorized as a macroautophagy; namely, damaged mitochondria are wrapped by a double-layer membrane structure (such as the double-layer membrane shed from the non-ribosomal attachment region of the rough endoplasmic reticulum) to form autophagosomes, and this step is followed by fusion of the outer membrane of the autophagosome with the lysosome membrane, entry of the substrate proteins into the lysosome, and ultimately, degradation of the substrates by various hydrolytic enzymes. Mitophagy plays an essential role in regulating the number and maintaining the normal function of mitochondria in the cell. Recent studies have revealed that mitophagy also plays an essential pathophysiological role in the development and replication of viruses, as will be reviewed here.

### 1.2. Activation Pathway of Mitophagy

The classic pathways that mediate mitophagy include the PINK1–Parkin and receptor-mediated pathways. The initiation of mitophagy is divided into ubiquitin-dependent and non-ubiquitin-dependent pathways depending on whether ubiquitin is involved or not (Figure 1).

#### 1.2.1. PINK1/Parkin Coordinate to Regulate Mitophagy

PTEN-induced putative kinase 1 (PINK1)/Parkin-driven mitophagy is the best characterized type of mitophagy. Recent studies have shown that ubiquitin-dependent mitophagy pathways are mainly regulated by PINK1/Parkin [1,2]. During exposure to stress conditions such as hypoxia, damaged mitochondria first undergo mitochondrial division mediated by dynamin-related protein 1 (Drp1), which results in the separation of the damaged and functional parts [3]. The damaged part of the mitochondria is unable to maintain a normal transmembrane potential; subsequently, the organelle membrane is depolarized and the potential decreases, resulting in damage to the transport channel of PINK1. Damage to mitochondria blocks PINK1 translocation and the protein is stabilized on the mitochondrial outer membrane. PINK1 recruits the cytoplasmic E3 ubiquitin ligase Parkin and enhances its ubiquitin ligase activity by phosphorylating its ubiquitin structure at Ser65 [4]. The phosphorylated Parkin then ubiquitinates mitochondrial outer membrane proteins, including voltage-dependent anion channel 1 (VDAC1), mitochondrial fusion protein 1 (MFN1), mitochondrial fusion protein 2 (MFN2), and Ras homologous gene family member C1 [5]. The ubiquitinated outer mitochondrial membrane proteins then interact with autophagy adaptors and microtubule-associated protein light chain 3 (LC3) through the ubiquitin-binding domain (UBD) and LC3 interacting region (LIR), respectively—thereby initiating mitophagy [6]. Nuclear dot protein 52 (NDP52) and optineurin (OPTN) have been identified as the central receptors for PINK1–Parkin-mediated mitophagy [7].

#### 1.2.2. PINK1 Mediates Parkin-Independent Mitophagy

PINK1 also recruits NDP52 and OPTN to the mitochondria and directly activates mitophagy, which is not mediated by Parkin [8]. Once recruited to the mitochondria, NDP52 and OPTN recruit the autophagy factors Unc-51-like kinase 1 (ULK1), double FYVE-containing protein 1 (DFCP1), and WD repeat domain phosphoinositide-interacting protein 1 (WIPI1) to the lesions near the mitochondria, which perform a precise identification function [7]. In cancer cells, the E3 ubiquitin ligase ARIH1/HHARI induces PINK1-dependent mitophagy through a mechanism independent of Parkin [9]. A study of mouse models of neurodegenerative diseases revealed that PINK1 knockout, instead of Parkin knockout, results in a continuous neurodegenerative disease—indicating that other PINK1-dependent phosphorylation sites might compensate for the role of Parkin in mitophagy [10].

#### 1.2.3. Autophagy Receptor-Mediated Mitophagy

The mitophagy signaling pathway also includes non-PINK1-dependent but receptor-mediated signaling, which is mainly mediated by autophagy receptors such as BCL2 interacting protein 3 like protein (BNIP3 L/NIX), FUN14 domain containing 1 protein (FUNDC1), and prohibitin 2 (PHB2) [11,12]. Unlike PINK1/Parkin-mediated ubiquitination of mitophagy, the receptors contain LIR regions on the outer mitochondrial membrane that directly bind to LC3 to initiate mitophagy without ubiquitination [13]. The LIR region interacts with molecules of the ATG8 family involved in autophagy, which specifically recognizes and transports autophagy substrates. Autophagy receptors are divided into two types: the first type contains ubiquitin-binding domains such as p62 protein, OPTN and nuclear dot protein 52 (NDP52), and the other type does not contain ubiquitin-binding domains, including ubiquitin ligase C (c-Cbl), nuclear receptor coactivator 4 (NCOA4), NIP3-like protein X (NIX), FUN14 domain-related protein 1 (FUNDC1), and starch binding domain-related protein 1 (STBD1) [14].

Moreover, autophagy substrates are recognized and bound by autophagy receptors in a ubiquitin-binding domain (UBA)-dependent or UBA-independent manner [15]. At the same time, autophagy substrates are anchored on the autophagosome membrane through LIR and then degraded in the lysosome. Different connection modes of the ubiquitin chain to some extent regulate the progression of mitophagy. Among them, lysine 27 (K27), K48, and K63-connected forms of polyubiquitin chains have been implicated in the degradation of autophagy substrates through the autophagy receptor p62 [16,17]. During mitophagy, receptors containing LIR regions that are located on the outer mitochondrial membrane directly bind to LC3 to initiate mitophagy without ubiquitination. Of course, these two mitophagy pathways do not exist separately. In the initiating stage of mitophagy, the ubiquitin kinase PINK1 phosphorylates ubiquitin to activate the ubiquitin ligase Parkin, which constructs the ubiquitin chain on the outer mitochondrial membrane protein and recruits autophagy receptors [18].

## 2. Mitophagy and Viral Infection

Mitophagy not only controls the quality of mitochondria but also has an essential function in viral infection [19]. As an immunoprotective mechanism of the body, mitophagy might play a role in destroying viruses. However, some viruses have evolved to escape or to utilize the abilities of mitophagy to facilitate their replication.

### 2.1. Viruses Inhibit Mitophagy

Studies on viruses and mitophagy have revealed that only a few viruses inhibit mitophagy flux. HIV infection promotes mitochondrial damage and alters mitochondrial dynamics. Furthermore, HIV single-stranded RNA (ssRNA) and the components gp120 and Tat impair mitophagy flux—thereby inhibiting mitophagy [20]. In addition, the activation of mitophagy impairs cell death caused by HIV-1 infection in astrocytes [21]. HCV induces mitophagy through the PINK–Parkin pathway, and HCV NS5A promotes ROS generation and then activates mitophagy [22]. However, the HCV core protein interacts with Parkin to rescue HCV-induced mitophagy by suppressing the translocation of Parkin to mitochondria [9].

### 2.2. Viruses Induce and Hijack Mitophagy

Some viruses have evolved immune evasion strategies to prevent them from being eliminated and to achieve persistent infection by directly inducing and using mitophagy at different stages (Figure 2). In the context of events involved in virus-induced mitophagy, hepatitis B virus (HBV), hepatitis C virus (HCV), Coxsackievirus group B (CVB), Venezuelan equine encephalitis virus (VEEV), classical swine fever virus (CSFV), and porcine reproductive and respiratory syndrome virus (PRRSV) trigger mitochondrial fission and subsequently induce Parkin-dependent mitophagy [12,23,24,25,26,27,28]. Accordingly, mitophagy promotes the replication of the aforementioned viruses. Mitophagy not only provides viruses a platform (autophagosome) for the assembly of replication complexes but also prevents viruses from being detected by innate immunity.

### 2.3. Mechanism through Which Viruses Induce Mitophagy

Different viruses induce mitophagy via distinct mechanisms. Transmissible gastroenteritis virus (TGEV) upregulates PARK7/DJ-1 (encoding a causative gene of familial Parkinson’s disease) and induces mitophagy to contribute to viral replication [29]. Studies have also shown that dengue virus (DENV) and pseudorabies virus (PRV) directly disrupt the dynamic changes in mitochondrial membranes and inhibit mitochondrial protein expression to induce mitochondrial damage and mitophagy [30,31]. Venezuelan equine encephalitis virus (VEEV) infection of human astrocytoma cells (U87MG cells) and African green monkey kidney cells induces mitophagy through the PINK1–Parkin pathway because Parkin is abundantly expressed on mitochondrial division fragments [32]. Similar findings have also been observed in CVB and CSFV-infected Vero cells and PK-15 cells [25,27].

Viruses also induce mitophagy via their viral proteins (Table 1). The influenza A virus (IAV) M2 protein interacts with LC3 through the LIR domain to recruit LC3 and form autophagosomes in HCT116 and A549 cells, and its PB1-F2 protein interacts with the Tu translation elongation factor, mitochondrial (TUFM) to induce mitophagy. The human parainfluenza virus (HPIV) M protein interacts with TUFM on the mitochondria, and recruits LC3 to form autophagosomes and facilitate viral replication [33,34]. DENV nonstructural protein 4B (NS4B) promotes mitochondrial elongation by inactivating the fission factor DRP1 and then induces mitophagy to stimulate DENV infection [35]. HCV nonstructural protein 5A (NS5A) induces mitophagy by triggering Parkin translocation to mitochondria in hepatoma cells. In the presence of bafilomycin A1 (BafA1), NS5A induces the accumulation of LC3 and a time-dependent decrease in p62 protein levels. Interestingly, the inhibition of reactive oxygen species (ROS) production attenuates NS5A-induced mitophagy. Severe acute respiratory syndrome coronavirus 2 (SARS-CoV-2) can encode open reading frame-9b (ORF-9b), which is localized in mitochondria and causes mitochondrial elongation; this elongation process subsequently expedites mitophagy to promote SARS-CoV-2 replication [36]. These studies indicate that some viruses have evolved different strategies to modulate mitophagy by mimicking short linear protein–protein interaction motifs in the host. These strategies lead to persistent virus infection by increasing the stability of progeny viruses and evading host immunity.

## 3. How Does Mitophagy Affect Viral Replication?

### 3.1. Mitophagy Affects Viral Replication by Regulating Immune Responses

Mitochondria play an important role in energy metabolism and the immune response, and the finding that mitophagy plays an important role in innate immunity is not surprising [37]. Mitophagy has been documented to negatively regulate innate immunity through signaling molecules recruited to the mitochondrial membrane, and this mechanism depends on the inner mitochondrial membrane potential (Δ*Ψm*) [38].

IAV infection induces mitophagy, which is essential for the cleavage of damaged mitochondria (Figure 3). Receptor-interacting protein kinase 2 (RIPK2) has an essential function in IAV-induced mitophagy. Deletion of RIPK2 increases the susceptibility of mice infected with IAV. IAV-infected RIPK2^-/-^ cells do not undergo mitophagy and exhibit increased mitochondrial production of superoxide and accumulation of damaged mitochondria, which negatively regulate NLRP3 inflammasome activation and IL-18 production (Figure 2) [39]. Moreover, the PB1-F2 protein derived from IAV polymerase basic protein 1 (PB1) is completely translocated into the mitochondrial inner membrane space via Tom20 channels, which leads to a reduction in Δ*Ψm*. The accumulation of PB1-F2 accelerates mitochondrial fission, and the attenuation of Δ*Ψm* inhibits the RIG-I signaling pathway and NLRP3 inflammasome activation. PB1-F2 translocation to the mitochondria is strongly associated with impaired cellular innate immune function. In addition, PB1-F2 functions as an autophagy receptor and simultaneously interacts with LC3 and TUFM domain I to mediate mitophagy. Notably, the C-terminal LIR motif of the PB1-F2 protein has been suggested to be a critical region for its binding to LC3 and the induction of mitophagy. Further studies have revealed that PB1-F2 antagonizes innate immune responses through a mechanism dependent on mitophagy. PB1-F2 LIR mutants decrease the negative regulation of PB1-F2 on type I IFN production and the degradation of mitochondrial antiviral signaling proteins (MAVs) [40]. Taken together, the results indicate that PB1-F2-induced mitophagy is closely correlated with impaired cellular innate immunity.

Mitophagy is critical for the degradation of MAVs. The IAV M2 protein possesses proton-selective ion channel activity and thus targets mitochondria upon IAV infection. This protein differentially regulates the expression of mitochondrial fission proteins (DNM1 L, MFF, and FIS1) and fusion proteins (Mfn1, Mfn2, and OPA1) to promote mitochondrial fusion. Furthermore, M2 protein expression increases mitochondrial numbers by facilitating the expression of proteins that mediate mitochondrial biogenesis. Significantly, the embodiment of the aforementioned functions of the M2 protein relies on its ion channel activity [41]. Unlike the PB1-F2 proteins, the M2 protein antagonizes the autophagy process to enhance host antiviral innate immune responses, and its mechanism for inhibiting autophagy is to induce PI3K–Akt–mTOR-dependent autophagosome formation while inhibiting autolysosome formation. Furthermore, M2-induced ion channel activity-dependent Ca^2+^ and ROS production is involved in M2-induced autophagy. On the one hand, the M2 protein increases ROS-dependent MAV aggregate formation through a mechanism dependent on its ion channel activity. On the other hand, the M2 protein antagonizes autophagy and competes with the autophagy genes ATG5 and LC3 for binding to MAVs, which results in the sequestration of proteins to prevent the formation of ATG5–MAV and LC3–MAV complexes, decreases in the degradation of MAVs aggregates, and thus the enhancement of MAV-mediated innate immune responses. This study provides novel insights into the interactions of IAV M2 with PB1-F2 proteins and host innate immunity, and thus provides a potential target for developing novel antiviral therapeutics for influenza virus infection.

In addition, our previous study revealed that the number of mitochondria encapsulated by autophagy-like vesicles is significantly increased in PK-15 and 3D4/2 cells infected with CSFV, which suggests that CSFV induces abnormalities in mitochondrial morphology and mitophagy [25]. Furthermore, the autophagy inhibitor 3-methyladenine (3-MA) and BafA1 (which inhibits the binding of autophagosomes and lysosomes) significantly decrease the amount of mitochondrial matrix protein in CSFV-infected cells, which indicates that CSFV infection induces complete mitophagy [25]. CSFV infection promotes the fusion of mitochondria and lysosomes and the expression of PINK1, Parkin, and LC3. Interestingly, as one of the key enzymes in glycolysis and metabolism, lactate dehydrogenase-B (LDHB) also plays a vital role in mitophagy. Fan et al. found that CSFV nonstructural protein (NS3) interacts with LDHB. CSFV infection induces mitophagy by inhibiting LDHB and promoting the ubiquitination of the Mfn2 protein. Additionally, LDHB-mediated mitophagy inhibits the activation of the NF-κB signaling pathway and decreases the release of the cytokines IFN-α, IFN-β, and TNF, which results in the promotion of the replication CSFV [42].

Mitophagy mitigates antiviral immune responses to promote the replication of multiple viruses. Attenuated measles virus (Mev) of the Edmonston strain (MV-Edm) plays a vital role in non-small-cell lung cancer cells (NSCLC). MV-Edm uses DDX58/RIG-I-like receptor (RLR)-mediated mitophagy, and this mitophagy promotes viral replication and inhibits the production of type I IFN through a process regulated by RLRs. MV-Edm also triggers p62-mediated mitophagy to result in decreased mitochondrial membrane potential-binding MAVs, and thereby weakens the innate immune response [43]. HPIV3 matrix protein is transferred to the mitochondria and induces mitophagy by interacting with the TUFM. M protein-induced mitophagy interferes with the RIG-I signaling pathway to inhibit the type I IFN response [44]. BK-polyomavirus (BKPyV) disrupts the mitochondrial network and mitochondrial membrane proteins. BKPyV promotes p62-mediated mitophagy by expressing a 66 aa-long agnoprotein during late replication, and this agnoprotein impairs the translocation of nuclear IRF-3 and decreases the expression of IFN-α and IFN-β to facilitate virus propagation [45].

### 3.2. Mitophagy Affects Viral Replication by Inducing Apoptosis

Apoptosis refers to a kind of programmed cell death and is another crucial process through which cells resist viral infection [46]. Apoptosis limits the replication and transmission of a virus by destroying the infected cell. A recent study has reported the relationship between mitophagy and apoptosis in cells infected with viruses [47]. TGEV induces mitophagy to eliminate ROS and alleviate apoptosis, subsequently enhancing TGEV infection in porcine epithelial cells [29]. However, CSFV induces mitochondrial fission and mitophagy to inhibit host cell apoptosis and ensure persistent infection [25]. PRRSV increases ROS levels to stimulate mitochondrial fission and mitophagy, which results in the attenuation of apoptosis in Marc145 cells and the promotion of PRRSV replication [28].

### 3.3. Mitophagy Affects Viral Replication through Autophagy Receptors

The vital role of autophagy receptors in antiviral research has received increasing attention in recent years. In CVB3 replication, the autophagy receptors p62 and NDP52 exert opposite regulatory effects on CVB3. Downregulation of the p62 gene leads to increased production of viral proteins and viral titer, whereas the inhibition of NDP52 expression results in significant inhibition of viral growth [48,49]. Both p62 and NDP52 directly interact with the viral capsid protein VP1 and promote VP1 ubiquitination, and as a result, these proteins aid autophagic vesicles in phagocytosing the virus [49]. NDP52 promotes the release of type I IFNs through MAVs, which exert antiviral effects. In RNA virus-infected cells, OPTN—which is located on the surface of the Golgi apparatus—interacts with polyubiquitinated TANK binding kinase 1 (TBK1) through its UBD and K63-ubiquitin chains. OPTN causes trans-autophosphorylation of the kinase and subsequently induces the phosphorylation of IRF3 and the production of type I IFNs [50]. ATG5 expressed in neurons protects the mouse central nervous system from the Sindbis virus (SINV), which promotes the expression of the autophagy receptor p62. SINV capsid protein interacts with p62 and targets it for autophagy-mediated degradation through p62 [51]. During IAV infection, polymerase basic protein 1 (PB1)-F2 plays an essential role in modulating virus virulence by inducing immune cell apoptosis and increasing verification. Olivier Leymarie et al. found that the IAV protein PB1-F2 interacts with NDP52 to regulate innate immunity by activating NF-κB and type I IFN-related pathways [52]. This activation is regulated by the upstream protein TRAF6. The NF-κB reporter experiment further revealed that the co-expression of NDP52, MAVS, and PB1-F2 enhances the inflammatory response, which indicates that NDP52 participates in regulating influenza viral replication through the PB1-F2 protein. Mev infection induces complete autophagy and thereby promotes virus replication. Similarly, NDP52 binds to Mev-C and Mev-V proteins that promote virus replication. A previous study found that both NDP52 and OPTN promote the maturation of autophagic vesicles [52]. NDP52 inhibition significantly reduces the replication of Mev, but does not affect the expression of the viral structural proteins Mev-N and Mev-P. Instead, NDP52 inhibition affects the replication of the virus by interfering with the synthesis of viral proteins [53]. Interestingly, NDP52 inhibits the NF-kB signaling pathway, but the effect of NDP52 on Mev does not depend on the NF-κB signaling pathway or T6BP [52]. Another study revealed that the Z matrix protein of the Lassa virus (LASV) and Mopeia virus (MOPV) also interacts with NDP52 [54].
cells-11-00711-t001_Table 1Table 1The function of mitophagy in viral infection.VirusInduction/Inhibition of MitophagyTargetMitophagy in VirusInfectionsViral Proteins Related to MitophagyReferencesHIVInhibition-Inhibitiongp120 and Tat[20]HBVInductionDNM1LPromotion-[23]HCVInductionDNM1L;ParkinPromotionNS5A[22,26]CVBInductionDNM1L;PINK1/Parkin;NDP52;P62Promotion/inhibitionVP1[27,49,55]VEEVInductionDNM1L;PINK1/ParkinPromotion-[32]DENVInductionDRP1PromotionNS4B[35]IAVInduction of incomplete autophagyLC3;NDP52;MAVSATG5PromotionM2;PB1-F2[33]MevInductionNDP52;-Mev-C;Mev-V[52,53]HPIVInductionTUFM;LC3PromotionM[33,34,44]MV-Edm (measles virus, Edmunston strain)Inductionp62Promotion-[43]BKPyVInductionp62;AgnoproteinPromotion-[45]CSFVInductionDNM1L; PINK1/ParkinPromotion-[24,25,42]PRVInhibition in Vero cells/Induction in N2a cells-Promotion-[30,31]PRRSVInduction of incomplete autophagyDNM1L;ROSPromotion-[28]TEGVInductionPARK7/DJ-1Promotion-[29]SARS-CoVInduction of mitochondrial elongation--ORF-9b[36]


## 4. Conclusions and Perspectives

Mitophagy not only strictly regulates the quality of mitochondria but also affects viral replication through different mechanisms. We systematically summarized the pathways involved in the initiation of mitophagy and elaborated on the role of mitophagy in virus infection.

At present, specific drugs for the treatment of most viruses are still unavailable, which poses a severe challenge to public safety and human health. The interaction between viruses and mitophagy remains to be systematically studied. Exploring the mechanisms of mitophagy in the context of virus infection and discovering the corresponding antiviral targets may provide a theoretical basis for the treatment and prevention of these viral infections.

## Figures and Tables

**Figure 1 cells-11-00711-f001:**
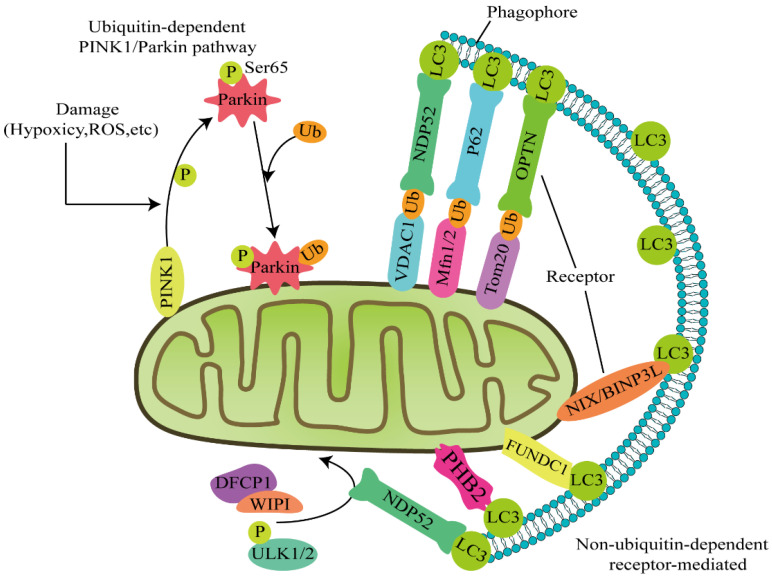
Two major pathways induce mitophagy: ubiquitin-dependent and non-ubiquitin-dependent pathways. Ubiquitin-dependent pathway: PINK1 is stabilized on the damaged mitochondrial outer membrane, recruits the cytoplasmic E3 ubiquitin ligase Parkin, and phosphorylates the ubiquitin structure of Parkin at Ser65. Phosphorylated Parkin ubiquitinates the mitochondrial outer membrane proteins VDAC1, Mfn1/2, and Tom20. At these sites, autophagy receptors such as NDP52 and OPTN initiate mitophagy by linking ubiquitinated mitochondrial outer membrane proteins to LC3. In addition, NDP52 recruits the autophagy factors ULK1, DFCP1, and WIPI to the lesions near the mitochondria for precise identification. Non-ubiquitin-dependent pathway: Autophagy receptors, such as BNIP3 L/NIX, FUNDC1, and PHB2, remain on the outer mitochondrial membrane and directly bind LC3 to initiate mitophagy without ubiquitination. In general, the two mitophagy pathways do not exist in isolation.

**Figure 2 cells-11-00711-f002:**
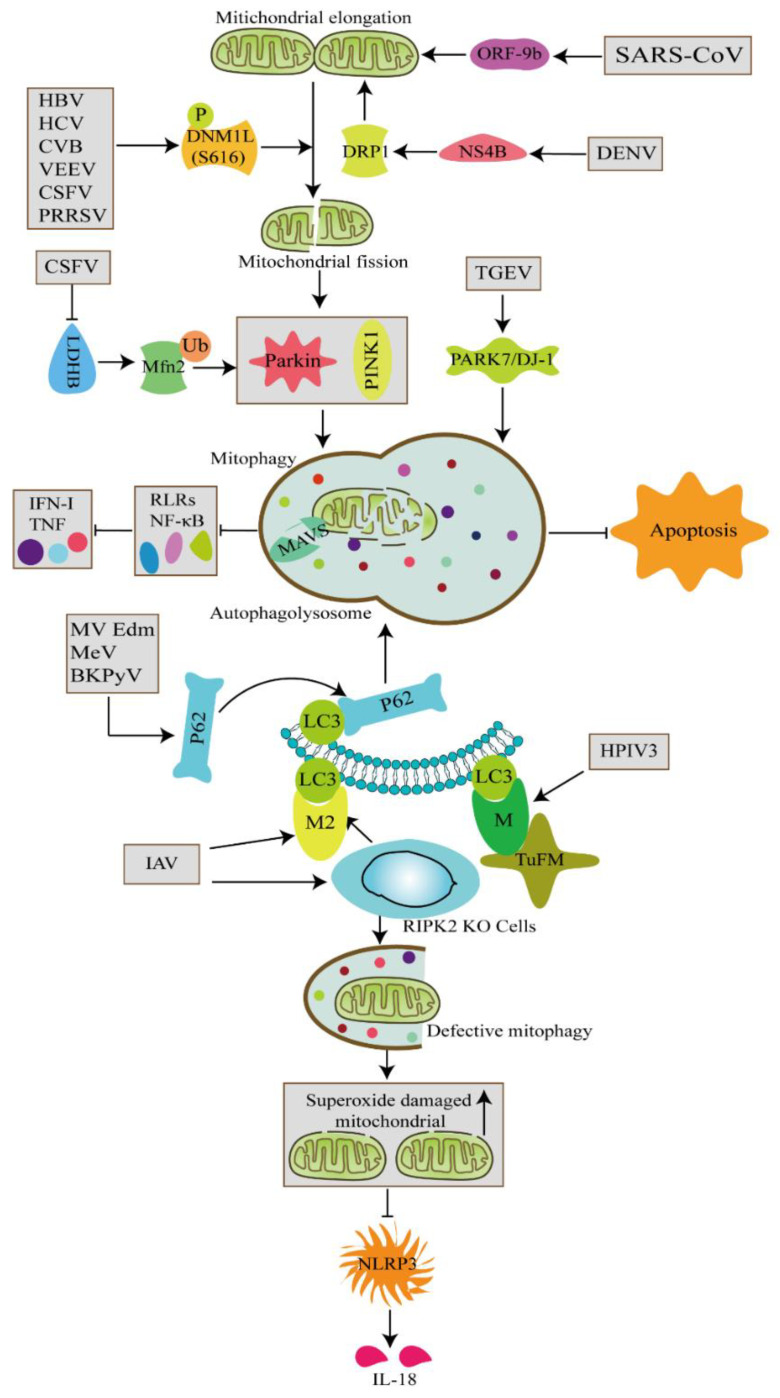
Viruses induce mitophagy at different stages and through viral proteins.

**Figure 3 cells-11-00711-f003:**
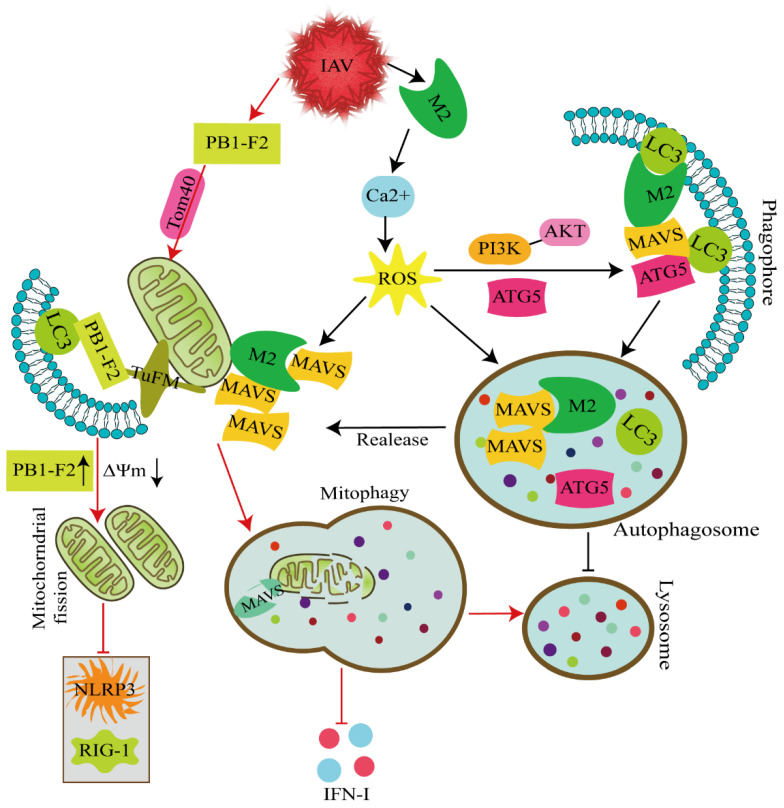
Molecular mechanism through which IAV regulates innate immunity via autophagy/mitophagy. IAV infection rapidly induces the host innate immune response through a variety of regulatory factors essential for pathogen clearance and host survival. PB1-F2 antagonizes innate immune responses by inducing mitophagy (as indicated by the red arrows). PB1-F2 completely translocates into the mitochondrial inner membrane space via Tom40 channels, which leads to a reduction in Δ*Ψm*. The accumulation of PB1-F2 accelerates mitochondrial fission, and the decrease in Δ*Ψm* inhibits the RIG-I signaling pathway and NLRP3 activation. In addition, PB1-F2 functions as an autophagy receptor protein and simultaneously interacts with LC3 and TUFM domain I to mediate mitophagy. PB1-F2 inhibits mitophagy-dependent type I IFN production by negotiating the sequestration of mitochondria (including MAVs) into autophagosomes. Unlike PB1-F2 proteins, the M2 protein antagonizes the autophagy process to enhance host antiviral innate immune responses. The M2 protein anchors to mitochondria, promotes mitochondrial fusion and increases the mitochondrial number. M2 induces PI3K–Akt–mTOR-dependent autophagosome formation but inhibits autolysosome formation. Furthermore, M2 increases ROS-dependent MAV aggregate formation through a mechanism dependent on its ion channel activity. Additionally, the M2 antagonist blocks the autophagy process of the interaction of MAVs with ATG5 or LC3 and, thereby reduces ATG5–MAV and LC3–MAV complex formation and decreases the degradation of MAV aggregates (as indicated by the black arrows).

## Data Availability

Not applicable.

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
