# Peer review of "The Role of Mitophagy in Viral Infection"

_cells, 2022, doi:10.3390/cells11040711_

Round 1

Reviewer 1 Report

This is a revised version of a previously submitted review article. The authors have improved language and style, but the structure and the flow of the text have not been altered. In its present form, the article is readable and in most parts well understandable. I do not believe that another round of review would make a difference.

There are still a few grammatical errors, e.g.

line 15/16:  mitopahgy is involved many - word missing

line 146: but also prevents viral from detection with innate immune - meaning unclear

line 172/173: can encodeencodes

line 298: interacts polyubiquitinated - word missing

line 334: all authors contributed to the the conception and design of the experiments - which experiments are meant here? This is a review article, which does not present any new experimental data. This statement does not make sense.

Author Response

Dear reviewer 1:

Thank you very much for your valuable comments. We have made relevant changes based on your suggestions. The details are as follows.

  1. There are still a few grammatical errors, e.g.
  2. line 15/16:  mitopahgy is involved many - word missing.

It has been changed into " mitopahgy is involved in many…"

  1. line 146: but also prevents viral from detection with innate immune - meaning unclear.

It has been changed into " but also prevents viruses from being detection with innate immune "

  1. line 172/173: can encodeencodes.

It has been changed into " can encode "

  1. line 298: interacts polyubiquitinated - word missing.

It has been changed into "interacts with polyubiquitinated TANK binding kinase 1 (TBK1) through its UBD and K63-ubiquitin chains".

  1. line 334: all authors contributed to the the conception and design of the experiments - which experiments are meant here? This is a review article, which does not present any new experimental data. This statement does not make sense.

It has been replaced by "All authors contributed to the conception and design of the work"

We appreciate your warm work earnestly and hope that the correction will meet with approval.

Once again, thank you very much for your comments and suggestions.

Sincerely,

Sen Zeng

College of Veterinary Medicine

South China Agricultural University

No. 483, Wushan Road, Tianhe District, Guangzhou 510642, People’s Republic of China

Email: 13425951792@163.com

Reviewer 2 Report

This manuscript has undergone several revisions. Through the process so far, the readability of the text has been significantly improved.

As this reviewer pointed out at the 1st submission, Author Contributions is unclear. In the Author Contributions section, there is a sentence that does not fit the review (design of the experiments!) and needs to be removed (lines 333-334).

Lines 52-53; Citations for “Recent studies” are needed.

Lines 281-283; Since there is one citation, isn't this passage singular?

Lines 201-204; Citations for “the report and studies” are needed.

Lines 247-250; Citations for “our previous study” are needed.

Author Response

Dear reviewer 2:

Thank you very much for your valuable comments on our manuscript. We have made relevant changes based on your suggestions. The details are as follows.

  1. As this reviewer pointed out at the 1st submission, Author Contributions is unclear. In the Author Contributions section, there is a sentence that does not fit the review (design of the experiments!) and needs to be removed (lines 333-334).

We are very sorry for our inaccurate description, considering the Reviewer’s suggestion, it has been replaced by "All authors contributed to the conception and design of the work”.

  1. Lines 52-53; Citations for “Recent studies” are needed.

Considering the Reviewer’s suggestion, we have added references at corresponding positions".

  1. Lines 281-283; Since there is one citation, isn't this passage singular?

As the Reviewer’s suggestion, it has been replaced by “A recent study has reported…”.

  1. Lines 201-204; Citations for “the report and studies” are needed.

Considering the Reviewer’s suggestion, we have added a reference at corresponding positions".

  1. Lines 247-250; Citations for “our previous study” are needed.

Considering the Reviewer’s suggestion, we have added a reference at corresponding positions".

We appreciate your warm work earnestly and hope that the correction will meet with approval.

Once again, thank you very much for your comments and suggestions.

Sincerely,

Sen Zeng

College of Veterinary Medicine

South China Agricultural University

No. 483, Wushan Road, Tianhe District, Guangzhou 510642, People’s Republic of China

Email: 13425951792@163.com

Reviewer 3 Report

The manuscript requires editing for English.

Author Response

Dear reviewer 3:

Thank you very much for your valuable suggestions on our manuscript. We have revised the grammar of the manuscript. We believe that the readability of this manuscript has been greatly improved. are as follows. You can see the details in the resubmitted manuscript.

  1. “The manuscript requires editing for English.“

As Reviewer suggested, we have made a thorough revision regarding the use of the English language and grammar in the legend. These changes will not influence the content and framework of the paper. And here we have not listed all of the changes but marked them in red in the revised paper.

We appreciate your warm work earnestly and hope that the correction will meet with approval.

Once again, thank you very much for your comments and suggestions.

Sincerely,

Sen Zeng

College of Veterinary Medicine

South China Agricultural University

No. 483, Wushan Road, Tianhe District, Guangzhou 510642, People’s Republic of China

Email: 13425951792@163.com